# Diverse Sphingolipid Species Harbor Different Effects on Ire1 Clustering

**DOI:** 10.3390/ijms232012130

**Published:** 2022-10-12

**Authors:** Mark A. Bieniawski, Kofi L. P. Stevens, Christopher M. Witham, Robert F. L. Steuart, Vytas A. Bankaitis, Carl J. Mousley

**Affiliations:** 1Curtin Medical School, Faculty of Health Sciences, Curtin University, Bentley, WA 6102, Australia; 2Curtin Health Innovation Research Institute, Faculty of Health Sciences, Curtin University, Bentley, WA 6102, Australia; 3Department of Molecular and Cellular Medicine, Texas A&M Health Sciences Center, Texas A&M University, College Station, TX 77843-1114, USA; 4Department of Biochemistry and Biophysics, Texas A&M University, College Station, TX 77843-1114, USA; 5Department of Chemistry, Texas A&M University, College Station, TX 77843-1114, USA

**Keywords:** unfolded protein response (UPR), Ire1, sphingolipids, Kes1, Osh4

## Abstract

Endoplasmic reticulum (ER) function is dedicated to multiple essential processes in eukaryotes, including the processing of secretory proteins and the biogenesis of most membrane lipids. These roles implicate a heavy burden to the organelle, and it is thus prone to fluctuations in the homeostasis of molecules which govern these processes. The unfolded protein response (UPR) is a general ER stress response tasked with maintaining the ER for optimal function, mediated by the master activator Ire1. Ire1 is an ER transmembrane protein that initiates the UPR, forming characteristic oligomers in response to irregularities in luminal protein folding and in the membrane lipid environment. The role of lipids in regulating the UPR remains relatively obscure; however, recent research has revealed a potent role for sphingolipids in its activity. Here, we identify a major role for the oxysterol-binding protein Kes1, whose activity is of consequence to the sphingolipid profile in cells resulting in an inhibition of UPR activity. Using an mCherry-tagged derivative of Ire1, we observe that this occurs due to inhibition of Ire1 to form oligomers. Furthermore, we identify that a sphingolipid presence is required for Ire1 activity, and that specific sphingolipid profiles are of major consequence to Ire1 function. In addition, we highlight cases where Ire1 oligomerization is absent despite an active UPR, revealing a potential mechanism for UPR induction where Ire1 oligomerization is not necessary. This work provides a basis for the role of sphingolipids in controlling the UPR, where their metabolism harbors a crucial role in regulating its onset.

## 1. Introduction

Lipid metabolism is a tightly regulated aspect of eukaryotic cell function. Membrane-enclosed organelles rely on a precise lipid composition in order to function appropriately [1,2]. The purpose of this is to maintain the structure of the membrane in both its fluidity and curvature [1,2,3]. Additionally, several families of membrane lipids are involved in a host of signaling events that direct processes such as the transport of materials to and from the organelle, regulating activity within the organelle, and broadcasting the state of conditions within the organelle to the wider cell [1,2].

Sphingolipids are a family of membrane lipids that are seemingly synonymous with the term ‘bioactive lipids’ in the recent literature. This is not surprising given that many sphingolipid species are seen directly involved in fundamental processes such as cell growth, senescence, and intracellular trafficking [4,5,6,7,8]. The abundance of sphingolipids has a direct impact on how these processes are regulated, and cells will often change their cellular sphingolipid profile as a means to potentiate homeostatic mechanisms. For this reason, the processes which govern sphingolipid metabolism must be managed by the cell, as deviations from normal levels can lead to inappropriate activities.

One particular field where studies in sphingolipid bioactivity have emerged is in the regulation of the unfolded protein response (UPR) [9]. The UPR is a highly conserved ER homeostatic mechanism in eukaryotes, with a significant number of studies being performed using a *Saccharomyces cerevisiae* model. Although defined as a response to misfolded protein accumulation in the ER, the UPR is generally viewed as a process for optimizing a cell’s secretory capacity according to its needs, doing so by adjusting the ER environment [10]. This response primarily drives ER homeostasis by reducing the cell’s secretory activity, increasing ER-associated degradation (ERAD), and increasing ER chaperone expression [10,11,12,13,14]. Other distinct processes occur during the UPR outside protein quality control including those that modulate ER membrane lipid composition, suggesting the UPR also plays a role ER membrane homeostasis. This idea has been repeatedly supported in the literature, with lipid disequilibrium demonstrated to activate the UPR independently of protein misfolding [15,16,17].

The primary mechanism that cells use to control the UPR is the ER transmembrane protein Ire1 [13]. Ire1 detects unfolded proteins in the ER with a luminal domain, this causes Ire1 to form oligomers [18]. Oligomerization of Ire1 allows their cytoplasmic RNase domains to combine and form an active RNase domain [19]. This new domain specifically targets constitutively expressed mRNA for the Hac1 transcription factor (Xbp1 in mammals), performing a non-conventional mRNA splicing event to yield a functional *HAC1* transcript [20]. The newly synthesized Hac1 serves as a potent activator for UPR-related gene expression [21]. Under normal conditions, Ire1 is maintained in a monomeric state through association with the luminal HSP70 family protein Kar2 (BiP in mammals). During ER stress, activation of Ire1 requires disassociation of Kar2, which occurs through the preferential association Kar2 has with the accumulating misfolded proteins [22,23]. The now free luminal domain of Ire1 can interact with unfolded proteins via an MHC-like groove within the luminal domain. Coordinating this activity with other free Ire1 proteins allows the formation of higher order oligomers that naturally condense into clusters throughout the ER [19,24,25,26]. The formation of clusters coupled with an inherent function for unspliced mRNA transcripts to be directed to cluster sites suggest that these structures serve as the cells signaling hubs for UPR control [27]. This offers interesting prospects for research into how the UPR might be regulated surrounding the process of constructing Ire1 clusters.

Several studies investigating the lipid-dependent regulation of the UPR demonstrate that Ire1 harbors lipid-sensory activity within its transmembrane domain, which alone can induce UPR activation [16,28,29]. This is illustrated when cells grown in inositol-deplete media exhibit a lipid-dependent UPR activation, a result of reduced synthesis of inositol-containing lipid derivatives [29]. Of particular interest to this study is the role that sphingolipids play in UPR activity. Sphingolipids are peculiar in their widely variable impact on the UPR. Where models displaying a moderate increase in sphingolipid abundance tend to elevate UPR activity, a massively increased sphingolipid presence in cells conversely mutes the UPR [9,30]. The mode for this polarized action remains unclear, however this poses interesting questions as to how sphingolipid metabolism might be regulated in order to control the onset of the UPR.

Utilizing models of Kes1 hyperactivity, we establish a lipid signaling-dependent pathway for regulating the UPR by targeting Ire1 clustering. This is due in large part to the relative increase in abundance of sphingolipid species. We have shown that Ire1 clustering can be either induced or inhibited depending on the relative abundance of dihydro- or phyto- sphingolipid families. Moreover, we demonstrate that misfolding-induced Ire1 clustering is heavily dependent on the presence of sphingolipids in the ER membrane. This highlights a potent regulatory mechanism for UPR activation. As an addition, we demonstrate systems of ER stress independent from lipid dysregulation/misfolding which harbor potent UPR activity however Ire1 is not seen to form clusters. This reveals a striking potential mechanism for activating the UPR where Ire1 oligomerization is not necessary. Our work provides a basis for how cells can modulate the ER lipid environment in order to regulate the onset of the UPR.

## 2. Results

### 2.1. Kes1 Activity Inhibits UPR Activity and Attenuates Ire1 Clustering

We have previously shown that induction of the UPR is attenuated in mutants, specifically *sec14-1^ts^ tlg2∆*, in which the oxysterol binding protein related protein Kes1 is hyperactivated [30]. In this mutant ER stress is registered as *HAC1* mRNA is spliced by Ire1. Instead, the Hac1 protein is unable to transactivate expression of UPR target genes [30]. That *HAC1* mRNA is spliced in *sec14-1^ts^ tlg2∆* cells suggested Ire1 to be activated in this mutant [30]. It has been shown that upon activation, Ire1 oligomerizes to form clusters in the ER membrane [24,25]. As such we anticipated Ire1 to form clusters in *sec14-1^ts^ tlg2∆* cells. To investigate this, we expressed a functional Ire1 derivative tagged with mCherry in the cytosolic linker that tethers the kinase domain of Ire1 to the transmembrane domain [27,31]. We observed Ire1 to redistribute from diffuse ER membrane staining to large distinct foci after DTT treatment in both wildtype (wt) and *sec14-1^ts^* cells (Figure 1A,B). Yet, we were surprised to discover that Ire1 did not form clusters in *sec14-1^ts^ tlg2∆* cells grown in the presence of DTT (Figure 1A,B). However, the deletion of *KES1* in this strain (*sec14-1^ts^ tlg2∆ kes1∆*) allowed for the formation of clusters when cells were treated with DTT (Figure 1A,B). Interestingly, considering the constitutive induction of the UPR in these cells (Figure 1C), we were surprised to discover that Ire1 did not cluster in untreated cells. Additionally, upon treatment with DTT the proportion of clustering cells in *sec14-1^ts^ tlg2∆ kes1∆* is reduced by ~1.6-fold in comparison to wt and *sec14-1^ts^* cells (Figure 1A,B). Taken together, this suggested that Ire1p clustering, although sufficient, is not necessary for activation of the UPR.

Given the complex genetic background of those analyses described above, we investigated UPR activation and Ire1 clustering in wt cells harboring the doxycycline (dox) repressible YCp P_DOX_-*KES1* plasmid grown in the presence and absence of dox and treated with and without DTT. In this system, Kes1 hyperactivation is achieved by elevating *KES1* expression approximately six-fold [32]. Firstly, we wanted to verify that this system phenocopies the UPR dysfunction observed in the *sec14-1^ts^ tlg2Δ* mutant. For this we measured UPR induced gene expression by quantitative RT-PCR as Kes1 hyperactivation by this method has been shown to reduce protein synthesis. We observe a robust increase in the transcription of genes induced by the UPR in wt cells treated with DTT relative to mock (Figure 2A). An equivalent response was also detected in wt cells harboring P_DOX_-*KES1* grown in the presence of dox to repress *KES1* expression following DTT treatment (Figure 2A). We observed a modest increase in the expression of genes induced by the UPR upon dox withdrawal (Figure 2A), however, this was not elevated following DTT treatment (Figure 2A). Furthermore, this was not due to inhibition of *HAC1* mRNA splicing as *HAC1^I^* was generated in all cells treated with DTT.

Importantly, the overexpression of the non-functional *KES1^K109A^* mutant had no effect on the expression of genes induced by the UPR in DTT treated cells (Figure 2A). Together, this reaffirms that Kes1 is an antagonist of the UPR. Again, Ire1 formed large, distinct foci after DTT treatment in wt cells grown in the presence of dox (Figure 2B), indicating that dox did not interfere with Ire1 clustering. However, Ire1 cluster formation is reduced ~5.6-fold upon the overexpression of functional *KES1* following the withdrawal of dox and is completely inhibited upon overexpression of the hyperactive mutant *KES1^Y97F^* (Figure 2B,C). Overexpressing the inactive mutant *KES1^K109A^* displays similar cluster formation to that seen with an empty vector (Figure 2B,C). Taken together this demonstrates that Kes1 hyper-activity inhibits Ire1 clustering in the ER membrane.

### 2.2. Sphingolipid Species Harbor Differential Impacts on Ire1 Clustering Activity

Sphingolipids and their metabolites regulate a diverse portfolio of cellular activities. Their synthesis starts in the ER through the condensation of serine and palmitoyl-CoA by serine palmitoyl transferase to produce 3-ketodihydroxysphingosine [33], which is subsequently reduced by Tsc10 to give dihydrosphingosine (DHS) [34]. At this point, DHS can be hydroxylated by Sur2 to produce phytosphingosine (PHS) [35] and both constitute the scaffolds on which all other sphingolipids are formed, albeit unequally, as more than 90% of the sphingolipidome is derived from the PHS scaffold [36]. We have previously shown dihydro- and phyto- sphingosine and ceramide levels to be significantly elevated in cells whereby Kes1 is hyperactivated [30,32]. In *sec14-1^ts^ tlg2Δ* cells, dihydroceramide and phytoceramide levels are elevated 10-fold relative to wt cells [30], and these lipid species were elevated to similar levels in cells overexpressing either *KES1* or *KES1^Y97F^* [32]. Given this characteristic of Kes1 hyperactivity, we aimed to determine the role of cellular sphingolipid content on Ire1 cluster formation. To do so, we investigated whether the treatment of cells with either DHS or PHS prevented DTT-dependent Ire1 clustering. We found PHS to be an effective inhibitor of DTT-dependent Ire1 clustering in wt cells (Figure 3A,B). In contrast, we were surprised to find that the acute exposure of wt cells to DHS stimulated Ire1 clustering in the absence of DTT (Figure 3A,B). Taken together, this demonstrates that PHS and DHS have different effects on Ire1 clustering, whereby PHS ablates Ire1 clustering, whereas DHS potentiates Ire1 clustering.

### 2.3. Phytosphingosine Inhibits Ire1 Clustering in sec12-4^ts^ Mutants

We sought to identify a model of ER stress that induces rapid Ire1 clustering without the need for treatment with chemicals that disrupt protein folding. Sec12 is a guanine nucleotide exchange factor which is essential for the formation of ER-derived vesicles, facilitating subsequent progress of ER cargo throughout the secretory pathway [37,38]. Loss-of-function mutants of Sec12 demonstrate a failure to effectively cycle cargo out of the ER, resulting in accumulated cellular ER mass [38,39]. Utilizing a temperature-sensitive mutant of Sec12 (*sec12-4 ^ts^*), we were able to see that upon shifting cultures to a non-permissive temperature (37 °C) for 2 h, Ire1 clustering significantly increases relative to that seen at permissive temperature (25 °C) (Figure 4A,B).

We wanted to determine whether PHS could inhibit the Ire1 clustering that occurs in *sec12-4^ts^* cells upon shift to its non-permissive temperature. Cells were treated with 40 µM PHS for 4 h prior to temperature shift. A modest increase in Ire1 clustering was observed at 25 °C in both wt and *sec12-4 ^ts^* cells (Figure 4A,B). At 37 °C, Ire1 clustering is shown to be significantly reduced in PHS-treated *sec12-4 ^ts^* cells, similar to levels seen in wt (Figure 4A,B). Taken together, these data demonstrate that PHS treatment inhibits Ire1 clustering irrespective of the source of ER stress.

### 2.4. De Novo Sphingolipid Synthesis Is Required for Ire1 Clustering

DHS is the precursor to all sphingolipids in yeast [34,36]. Given our observation that DHS potentiates Ire1 clustering we hypothesized that the inhibition of DHS synthesis would ablate Ire1 clustering. For this, we performed treatments on wt cells with the serine palmitoyl transferase inhibitor myriocin and determined the proportion of cells with Ire1 clusters post-DTT treatment. Wt cells were treated overnight with or without 700 ng/mL myriocin prior to treatment with 5 mM DTT. Strikingly, an inhibitory concentration of myriocin (700 ng/mL) displayed a near-complete inhibition of Ire1 clustering (Figure 5A,B). These data suggest that DHS synthesis is required for Ire1 oligomerization in response to protein misfolding.

### 2.5. The UPR Can Be Active Independently from Ire1 Clustering

Earlier when investigating Ire1 clustering in the *sec14-1^ts^ tlg2∆ kes1∆* background, no clustering is observed despite these cells exhibiting a significant activation of the UPR. Since Ire1 clustering is a characteristic checkpoint for activating the UPR, this is a peculiar finding. Considering the complex genetic background of the cells analyzed above, we sought to identify other models for ER stress showing similar characteristics. Sss1 is an essential component of the ER translocon, with recent research highlighting a major role in stabilizing the gating of small molecules and ions between the cytosolic and ER luminal environments [40]. Temperature sensitive variants of *SSS1*, *sss1-6^ts^* and *sss1-7^ts^* show poor maintenance of ER homeostasis and thus have a constitutively active UPR at all temperatures (Figure 6A). Despite this, Ire1 does not relocalize into clusters even at non-permissive temperatures (Figure 6B,C). However, like for *sec14-1^ts^ tlg2∆ kes1∆*, Ire1 does form clusters when cells are exposed to DTT (Figure 6B,C). This demonstrates that Ire1 oligomerization, although characteristic, is not entirely necessary for maintaining activity of the UPR.

## 3. Discussion

In this study, we identified a novel lipid-based process for regulating UPR activity. Kes1 antagonizes activation of the UPR, suggesting Kes1 as a lipid sensor-based mechanism for UPR control. This is achieved through the role of Kes1 in coordinating cellular sphingolipid metabolism, where derivatives of sphingosine species perform as the major effectors. Ire1 serves as one target for this effect, where its capacity to form clusters is impacted by changes to the sphingolipid profile; consistent with previous reports of Ire1 activity being regulated in a lipid-dependent manner. This facet of UPR regulation can override Ire1 activity across sources of ER stress, thus highlighting the crucial role for lipid signaling in regulating the onset of the UPR.

The UPR is classically characterized to be activated via two routes: the accumulation of misfolded proteins in the ER lumen, and an aberrant ER lipid environment [11,12,15]. Although occurring in separate environments of the same organelle, either stress will produce a similar response, resulting in the upregulation of genes for both protein-misfold management and lipid metabolism. This is explained by the function of Ire1. As the master activator of the UPR in yeast, Ire1 is responsive to both forms of ER dyshomeostasis but provides only one signal output in the form of Hac1 expression [21]. Our data have shown that an elevated cellular sphingolipid presence can shift UPR control to be independent of ER stresses. Similarly, by acutely inhibiting the synthesis of sphingolipids, DTT-dependent Ire1 clustering is inhibited.

Sphingolipids are shown to have bipolar effects on the UPR depending on the context, however a biochemical understanding of this has remained obscure. During our experimentation, we utilized DHS and PHS to determine their effects on Ire1 clustering. In eukaryotes, DHS can be hydroxylated at C1 along its acyl chain to produce PHS [35]. Dihydro- and phyto- sphingosine derivatives are processed into mature sphingolipids, separating into two families [35]. Our data show that DHS opposes the role to PHS in Ire1 clustering, where cluster formation occurs regardless of DTT treatment. Given that PHS derivatives are more abundant than those of DHS in Kes1-hyperactive cells, it is likely that the cluster-promoting activity from dihydro-sphingolipids is being masked. However, this provides a basis for how changes in sphingolipid metabolism can produce dichotomous outcomes for UPR activity.

The transmembrane domain of Ire1 is shown to coordinate its activity in response to changes in the ER membrane lipid environment. Halbleib et al., 2017 [28] highlighted mutations within the transmembrane domain which significantly reduce lipid-dependent clustering, where the responsiveness of Ire1 to lipid bilayer stresses was disrupted [28]. The impacts of both DHS and PHS on Ire1 clustering are distinct despite their close relationship. Our data, although not showing a direct causality, allow us to speculate that Ire1 responds differently to these sphingolipid species due to differences in their respective physiochemical properties. This would be explained by the variable hydroxy- modification; where PHS-derived sphingolipids could disrupt the necessary interactions between the Ire1 transmembrane domain and ER membrane to form oligomers.

Our findings establish Kes1 activity as a potent regulator for UPR activation, adding to its already large panel of regulatory influence. Previously, Kes1 was shown to be a potent antagonist of Gcn4 dependent transactivation of the general amino acid control (GAAC) pathway [32]. Here, Kes1 hyperactivation results in the non-productive binding of enhancer elements of GAAC induced genes by Gcn4 via a mechanism dependent on sphingolipids and the cyclin dependent kinase 8 module of the large Srb-Mediator complex [32]. Gcn4 has been shown to be an essential co-regulator of the UPR in yeast, binding promoters of target genes alongside Hac1 to stimulate transcriptional induction [41]. In light of this, we conclude that the role of Kes1 as an UPR antagonist is the result of its ability to potently ablate Gcn4 activity. The mechanics behind Kes1-dependent regulation of sphingolipid metabolism are unclear, presenting an avenue in research for determining its role in how this is coordinated. The UPR remains at the heart of much up and coming research, particularly in how the response manifests in human disease. Being as lipid-dependent regulation of the UPR has remained rather enigmatic, our work has allowed us to build a more cohesive understanding of this, and how the UPR is integrated with other major cell functions.

Our additional findings also highlight that the UPR can be active despite the absence of the characteristic formation of Ire1 clusters. This has been observed in systems displaying ER stress outside those classically characterized to normally induce the UPR, namely in Sss1 mutants which destabilize small molecule homeostasis in the ER. This reveals a peculiar mechanism of regulating UPR activity where Ire1 clustering is not necessary. It has been proposed that activation of a weak, Ire1-dependent UPR initiated without the formation of Ire1 clusters is a less harmful and more favorable mechanism for the cell to manage prolonged ER stress [42]. When needed however, a switch to a more potent UPR is provided upon Ire1 clustering by a mechanism that requires recognition of lumenal unfolded/misfolded proteins by Ire1′s MHC-like groove and altered lipid content of the ER membrane. Regarding the latter, Schuldiner and colleagues have shown that heme-dependent sterol biosynthesis is critical for Ire1 cluster formation [27,31]. We suggest that the synthesis of dihydro- sphingolipid species should be considered in the same vein. Finally, we speculate that accumulation of phyto-/hydroxylated sphingolipids may be required to dampen the UPR. Here, as Ire1 monomers/dimers dissociate from clusters phyto- sphingolipids would prevent them from re-oligomerizing. Second, as reported in Mousley et al., 2008 [30], these sphingolipids would also attenuate Hac1 dependent transactivation of UPR genes. Thus, phyto-sphingolipids establish a poise in the ER that results in the attenuation of the UPR.

Taken together, this study suggests that the sphingolipid content of the ER dictates the potential for Ire1 to respond appropriately to stimuli within the ER and future work will provide functional insight into how these lipids affect Ire1 activity.

## 4. Materials and Methods

### 4.1. Yeast Strains and Media

*Saccharomyces cerevisiae* strains and plasmids used are listed in Appendix A, respectively. Yeast strains were grown routinely at 30 °C in minimal medium (0.67% yeast nitrogen base; YNB) with 2% D-glucose plus appropriate supplements for selective growth. Media components were all purchased from Formedium (Hunstanton, UK). For experiments involving temperature sensitive strains, cells were cultured for 2 h at 37 °C prior to testing. Cells were cultured in media containing 25 µM DHS or 45µM PHS with 0.05% TritonX-100 for 4 h prior to testing as indicated. DTT (Sigma, Aldrich, St. Louis, MO, USA) was used as 5 mM or 10 mM treatments for 2 h, and Doxycycline used at 10 µg/mL as indicated. Myriocin was sourced from Sigma.

### 4.2. Vector Construction

We required the selectable marker present in the P_DOX_-*KES1* series of vectors generated by Mousley et al. 2012 [32] to be switched from *URA3* to *HIS3*. Cells harboring P_DOX_-*KES1*, P_DOX_-*KES1^Y97F^* or P_DOX_-*KES1^K109A^* were transformed with pUH7 [43] cut with *Sma I* restriction endonuclease. Transformants were grown on minimal media containing 5′ fluoroorotic acid but lacking histidine. Plasmids were isolated from cells and re-transformed into wt yeast and the ability to grow on media lacking histidine but not uracil reassessed. The correct insertion of the *HIS3*-kanR fragment into the *URA3* gene of each plasmid was confirmed by restriction with *Stu I*.

### 4.3. Fluorescence Confocal Microscopy

Cultured cells containing pMS383 (Ire1-mCherry expression) [27] were harvested and fixed with 3.7% formaldehyde for 15 min and mounted on slides coated with concanavalin-A. Static images were then collected using an Ultraview Spinning Disk Confocal Microscope (Perkin Elmer Life Sciences). Cell counts were subsequently taken.

### 4.4. Reverse Transcription (RT)-PCR

Total RNA was isolated from cells and 1μg was used to template reverse transcription to generate cDNA (20 μL final volume). To analyze *HAC1*, *KAR2*, *ERO1*, *PDI1* or *ACT1* expression, PCR was performed using 1μL of cDNA fraction as template and specific oligonucleotides as primers. Products were quantified with ImageQuant (GE Healthcare Life Science).

### 4.5. β-Galactosidase Assays

β-Galactosidase assays were performed according to Tyson and Stirling 2000 [44]. Briefly, yeast cells were grown at 30 °C in minimal medium containing 2% glucose and appropriate supplements. Cultures were diluted to A_600 nm_ of 0.2 and grown for a further 4 h. Cells were isolated and resuspended in 2 mL of Z buffer (60 mM Na_2_HPO_4_, 40 mM NaH_2_PO_4_, 10 mM KCl, 10 mM MgSO_4_, 50 mM 2-mercaptoethanol, pH 7.0). Aliquots (0.8 mL) were collected, cells were permeabilized in 50 μL of 0.1% (*w*/*v*) SDS and 100 μL of CHCl_3_, and samples were equilibrated to 30 °C. Assays were initiated by addition of 160 μL of *o*-nitrophenyl-galactopyranoside (4 mg/mL stock solution in Z buffer) and incubated at 30 °C for 20 min. Reactions were terminated by addition of 400 μL of 1 M Na_2_CO_3_, pH 9.0, the OD_420 nm_ was measured, and LacZ activity (U) was calculated by multiplying OD_420 nm_/OD_600 nm_ by 1000. Three independent biological replicates and at least two technical replicates were performed.

## Figures and Tables

**Figure 1 ijms-23-12130-f001:**
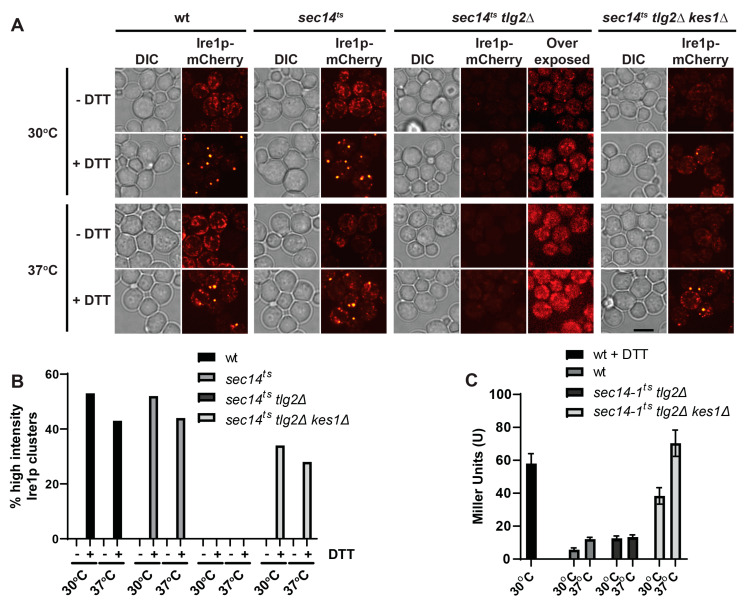
Kes1p hyperactivity attenuates UPR activity by attenuating Ire1 clustering. (**A**) Fluorescence microscopy images taken from wt (CTY182) *sec14^ts^* (CTY5-2), *sec14^ts^ tlg2Δ* (CTY1920), and *sec14^ts^ tlg2Δ kes1Δ* (CTY1958) expressing Ire1-mCherry. Cells were cultured at 30 °C and 37 °C for 2 h, followed by treatment with ±10 mM DTT. Over-exposed images of CTY1920 are shown to address low signal intensity. (**B**) Cell counts determining the proportions of cells containing high-intensity Ire1-mCherry clusters, taken from microscopy images described above. At least 150 cells were counted for each condition. (**C**) Wt (CTY182), *sec14^ts^ tlg2Δ* (CTY1920), and *sec14^ts^ tlg2Δ kes1Δ* (CTY1958) cells transformed with pJT30 (UPRE-LacZ) were grown in –Ura selective medium and β-Galactosidase activity determined. As a positive control, wild-type cells were treated with 5 mM DTT for 2 h.

**Figure 2 ijms-23-12130-f002:**
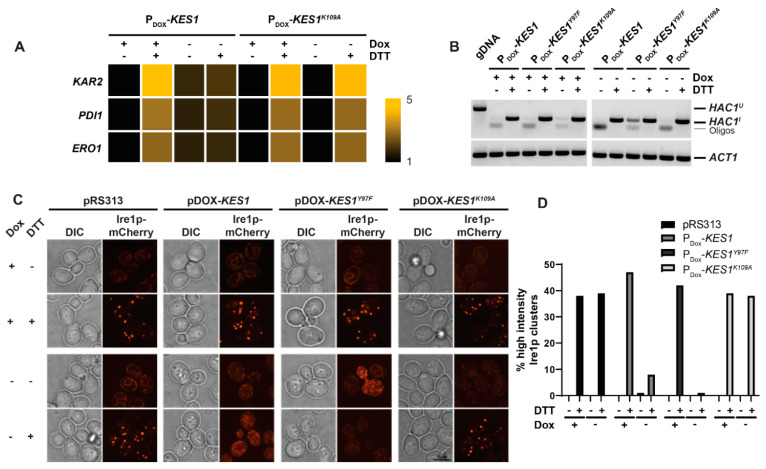
Kes1p hyperactivity attenuates the UPR and Ire1p clustering. (**A**) RT-PCR data of UPR-associated genes in wild-type cells containing vectors for doxycycline-repressible P_DOX_-*KES1*, P_DOX_-*KES1^Y97F^* and P_DOX_-*KES1^K109A^* overexpression. Cells were treated with ±5 mM DTT for 2 h prior to sample collection. (**B**) RT-PCR of *HAC1* in wild-type cells containing vectors for doxycycline-repressible P_DOX_-*KES1*, P_DOX_-*KES1^Y97F^* and P_DOX_-*KES1^K109A^* overexpression. Cells were treated with ±5 mM DTT for 2 h prior to sample collection. gDNA was used as a negative control for *HAC1* splicing and *ACT1* was used as RT-PCR control. (**C**) Fluorescence microscopy images taken of Ire1-mCherry from wild-type cells transformed to determine role of *KES1* overexpression of Ire1 clustering. Vectors used include pRS313 (empty *HIS3* vector), P_DOX_-*KES1*, P_DOX_-*KES1^Y97F^*, and P_DOX_-*KES1^K109A^*. Cells were cultured either in the presence or absence of 10 μg/mL doxycycline. (**D**) Cell counts determining the proportions of cells containing high-intensity Ire1-mCherry clusters (taken from microscopy images described above). At least 150 cells were counted for each condition.

**Figure 3 ijms-23-12130-f003:**
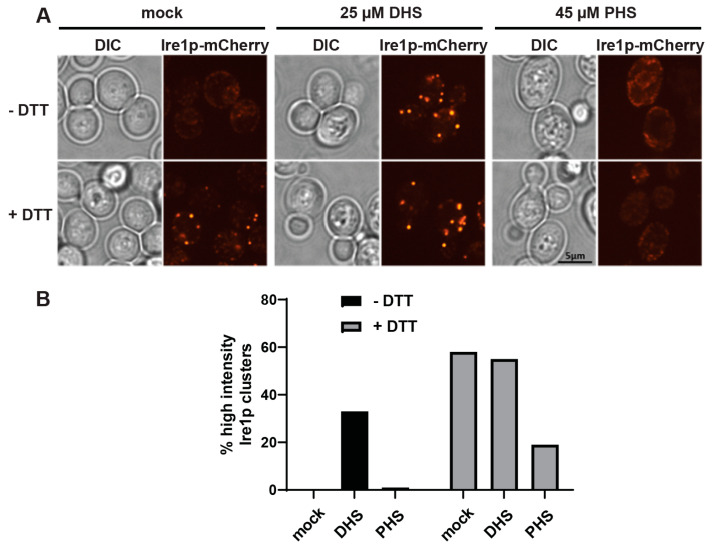
Ire1p clustering is sensitive to sphingolipids. (**A**) Fluorescence microscopy of wild-type cells expressing Ire1-mCherry treated with either 25µM DHS, or 45µM PHS, followed by treatment with ±5 mM DTT. (**B**) Cell counts determining the proportions of cells containing high-intensity Ire1-mCherry clusters (taken from microscopy images described above). At least 150 cells were counted for each condition.

**Figure 4 ijms-23-12130-f004:**
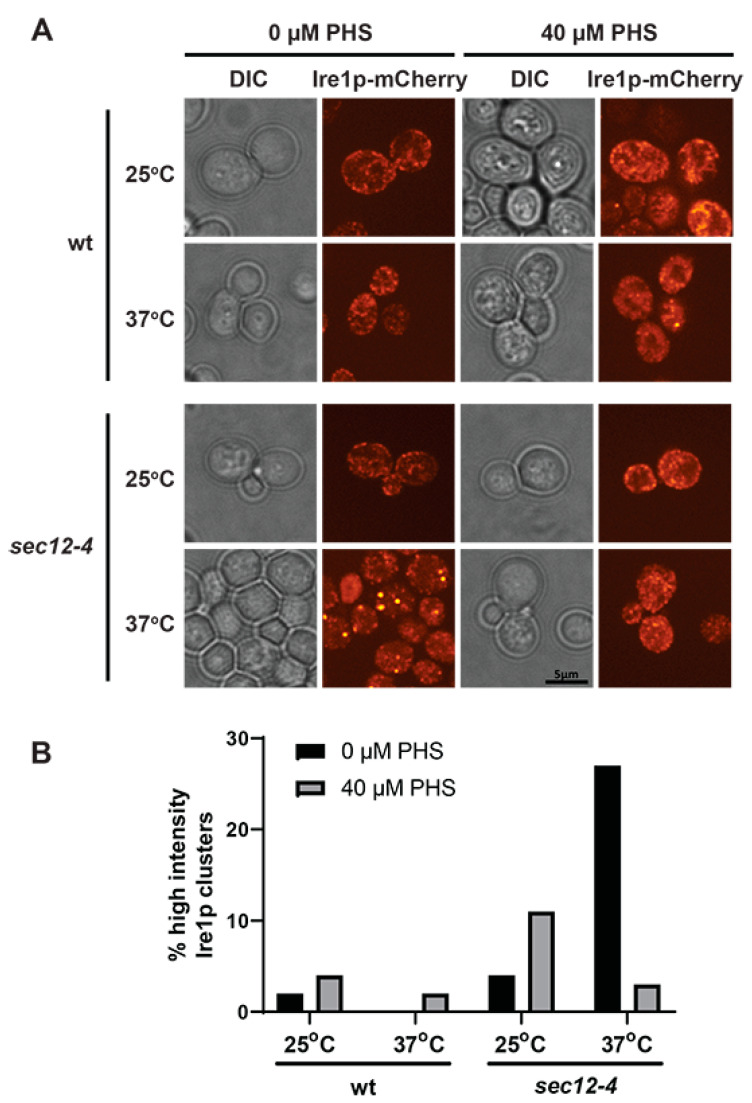
ER transport-derived Ire1 clustering is inhibited by phytosphingosine. (**A**) Fluorescence microscopy images of wt (CTY182) and *sec12-4^ts^* (CTY252) cells expressing Ire1-mCherry. Cells were cultured at 25 °C with 40 µM PHS 4 h, followed by shifting cultures to either 25 °C or 37 °C for 2 h. (**B**) Cell counts determining the proportions of cells containing high-intensity Ire1–mCherry clusters (taken from microscopy images described above). At least 150 cells were counted for each condition.

**Figure 5 ijms-23-12130-f005:**
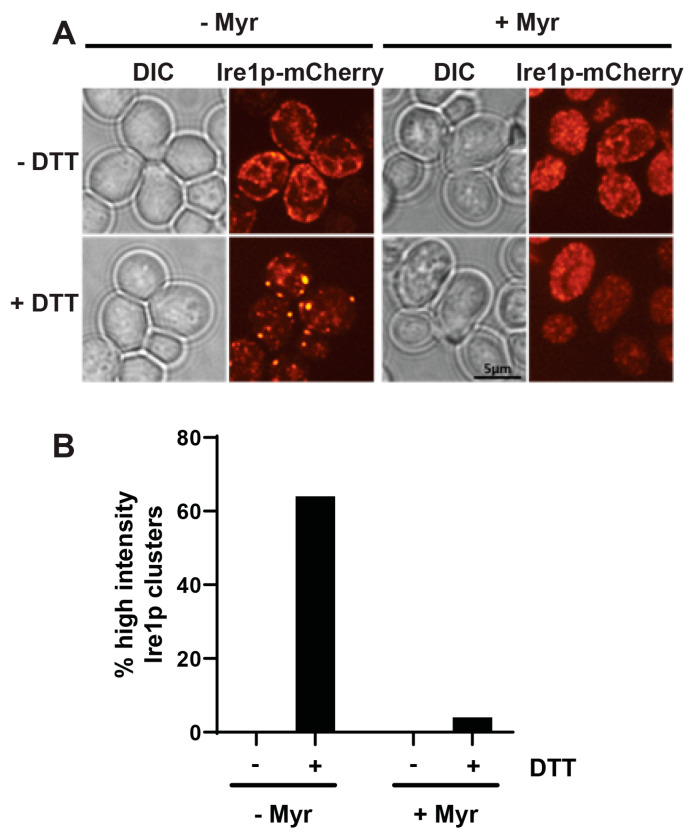
Ire1p clustering is sensitive to sphingolipids. (**A**) Fluorescence microscopy of wt cells expressing Ire1p-mCherry ±700 ng/mL myriocin. (**B**) Cell counts taken from fluorescence microscopy images of wt cells expressing Ire1-mCherry treated with ±700 ng/mL myriocin, followed by treatment with ±5 mM DTT. Proportions of high-intensity clustering cells were measured and at least 150 cells were counted for each condition.

**Figure 6 ijms-23-12130-f006:**
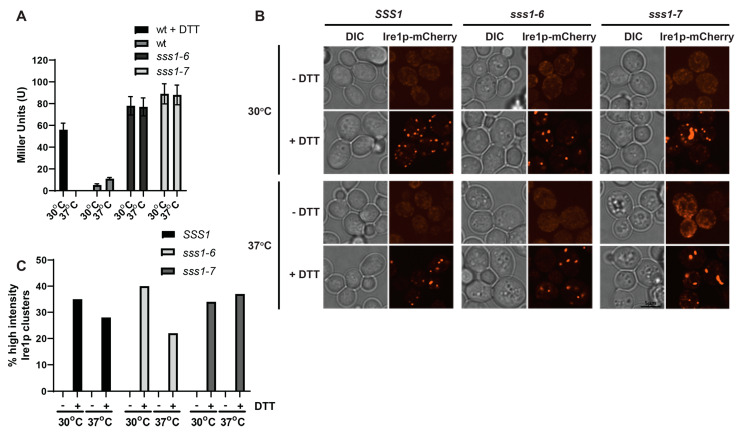
The UPR can be active despite the absence of Ire1 cluster formation. (**A**) Fluorescence microscopy images of wt, *sss1-6* and *sss1-7* cells expressing Ire1-mCherry. Cells were cultured at 30 °C and 37 °C for 2 h, followed by treatment with ±10 mM DTT. (**B**) Cell counts determining the proportions of cells containing high-intensity Ire1-mCherry clusters (taken from microscopy images described above). At least 150 cells were counted for each condition. (**C**) Wt, *sss1-6* and *sss1-7* cells transformed with pJT30 (UPRE-LacZ) were grown in –Ura selective medium and β-Galactosidase activity determined. As a positive control, wildtype cells were treated with 5 mM DTT for 2 h.

## Data Availability

Not applicable.

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
