# Peer review of "Diverse Sphingolipid Species Harbor Different Effects on Ire1 Clustering"

_ijms, 2022, doi:10.3390/ijms232012130_

Round 1

Reviewer 1 Report

The manuscript is well-written and scientifically sound. The study focuses on the identification of a new lipid-based process involving Kes1 in the unfolded protein response control.  This study highlights a crucial regulatory connection between the sphingolipid derivatives and the UPR. Multi-knockout mutants and drug inhibitory assays were the major tools that were used to investigate this Ire1/Kes1 control in UPR.  

Major concern: 

This manuscript lacks a study identifying the molecular mechanism by which Kes1 inhibits UPR activity and partially blocks Ire1 clustering!

Minor concern:

In Fig 1B color code for sec14 and sec14tlg2 seems to be similar

Author Response

We would like to thank reviewer 1 for taking the time to review our manuscript. Reviewer 1 indicated "This manuscript lacks a study identifying the molecular mechanism by which Kes1 inhibits UPR activity and partially blocks Ire1 clustering!" Addressing the latter point first. We note that sphingolipids, particularly phytosphingolipids, accumulate in all cases that Kes1 is hyperactivated. This study recapitulates the effect by feeding sphingolipids exogenously. Therefore, we have cut out the middle man, that is hyperactivation of Kes1, in this analysis focusing on the effect of sphingolipids themselves. We were simply using Kes1 as a tool to set the scence; that sphingolipids are impacting Ire1 clustering. We readily admit that we do not know the mechanim as to how sphingolipids accumulate upon Kes1 hyperactivation (stating this in the discussion) and suggest that this will provide the basis of future investigations

Regarding the former, i.e. "This manuscript lacks a study identifying the molecular mechanism by which Kes1 inhibits UPR activity". This is a very good question and one that hopefully I will answer sufficiently. We reported in Mousley et al., 2012 that Kes1 hyperactivation attenuates Gcn4 dependent activation of the general amino acid control (GAAC). Kes1 hyperactivation results in the non-productive binding of Gcn4 to enhancer elements of GAAC induced genes by a mechanism dependent on sphingolipids and the CDK8 module of the large Mediator complex. Gcn4 has also been shown to be an essential co-regulator of the UPR in yeast, binding promoters of target genes alongside Hac1 to stimulate transcriptional induction. In light of this, we conclude that the role of Kes1 as an UPR antagonist is the result of its ability to potently ablate Gcn4 activity.

We have added the following to the discussion:

'Previously, Kes1 was shown to be a potent antagonist of Gcn4 dependent transactivation of the general amino acid control (GAAC) pathway 32. Here Kes1 hyperactivation results in the non-productive binding of enhancer elements of GAAC induced genes by Gcn4 via a mechanism dependent on sphingolipids and the cyclin dependent kinase 8 module of the large Srb-Mediator complex 32. Gcn4 has been shown to be an essential co-regulator of the UPR in yeast, binding promoters of target genes alongside Hac1 to stimulate transcriptional induction 41. In light of this, we conclude that the role of Kes1 as an UPR antagonist is the result of its ability to potently ablate Gcn4 activity.'

Minor comment

"In Fig 1B color code for sec14 and sec14tlg2 seems to be similar" Thank you for indicating this to us. We have changed the colour code of sec14 tlg2 to a darker shade

We hope that this addresses the concerns of the reviewer.

Reviewer 2 Report

The paper in very well written and experiments are well exposed and designed. I think the paper should be published as it is

Author Response

We would like to thank reviewer 2 for taking the time to review our manuscript and indicating that it is of sufficient quality for publication in its current state. 

Round 2

Reviewer 1 Report

I appreciate authors taking time and addressing to Kes1 part, however exploring that mechanism would have opened the novelty of the manuscript. I hope you could identify the molecular mechanism your future studies.